# Ethyl Lauroyl Arginate, an Inherently Multicomponent Surfactant System

**DOI:** 10.3390/molecules26195894

**Published:** 2021-09-29

**Authors:** Agnieszka Czakaj, Ewelina Jarek, Marcel Krzan, Piotr Warszyński

**Affiliations:** Jerzy Haber Institute of Catalysis and Surface Chemistry, Polish Academy of Sciences, 30-239 Krakow, Poland; ewelina.jarek@ikifp.edu.pl (E.J.); marcel.krzan@ikifp.edu.pl (M.K.)

**Keywords:** ethyl lauroyl arginate (LAE), surface tension, surface activity, surface dilational elasticity, hydrolysis, dimerisation

## Abstract

Ethyl lauroyl arginate (LAE) is an amino acid-based cationic surfactant with low toxicity and antimicrobial activity. It is widely used as a food preservative and component for food packaging. When stored, LAE decomposes by hydrolysis into surface-active components Nα-lauroyl–l-arginine (LAS) or dodecanoic (lauric) acid. There are only a limited number of reports considering the mechanism of surface activity of LAE. Thus, we analysed the surface tension isotherm of LAE with analytical standard purity in relation to LAE after prolonged storage. We used quantum mechanical density functional theory (DFT) computations to determine the preferred hydrolysis path and discuss the possibility of forming highly surface-active heterodimers, LAE-dodecanoate anion, or LAE-LAS. Applying molecular dynamics simulations, we determined the stability of those dimers linked by electrostatic interactions and hydrogen bonds. We used the adsorption model of surfactant mixtures to successfully describe the experimental surface tension isotherms. The real part surface dilational modulus determined by the oscillation drop method follows a diffusional transport mechanism. However, the nonlinear response of the surface tension could be observed for LAE concentration close to and above Critical Micelle Concentration (CMC). Nonlinearity originates from the presence of micelles and the reorganisation of the interfacial layer.

## 1. Introduction

Ethyl lauroyl arginate (LAE) is an amino acid-based cationic surfactant synthesised from l-arginine, lauric acid and ethanol [1]. It has been approved and generally recognised as safe (GRAS) for some food and biomedical applications by the USA Food and Drug Administration (FDA) and the European Food Safety Agency (EFSA) [2,3]. Toxicological studies have demonstrated LAE’s low toxicity as it can be hydrolysed by chemical and metabolic pathways into components that are easily further metabolised [3,4]. Ethyl lauroyl arginate has strong antimicrobial activity against various microorganisms, including moulds, yeasts, Gram-positive, and Gram-negative bacteria. As a cationic surfactant, it can penetrate the bacterial cytoplasmic membrane that causes its deformation and the loss of cell viability [5,6,7]. There are numerous reports on the application of LAE as a food preservative and component for food packaging [2,8]. The interactions of ethyl lauroyl arginate with biopolymers were investigated using various physicochemical methods [2,9]. The results indicated strong electrostatic binding between LAE and anionic biopolymers leading to complex formation that can have implications for the formulation of delivery systems [10] or other industrial applications [11]. Strong binding of LAE to the surface of negatively charged nanoparticles can be used to modify their surface properties for particular applications. For example, ethyl lauroyl arginate was used to modify the cellulose nanocrystals (CNC) for the improvement of foamability and foam stability [12] or stability of emulsions [13]. Three types of LAE-CNC interactions were recognised: the electrostatic attraction at low surfactant concentrations, followed by the hydrophobic interaction, and polymer-induced micellisation [14].

LAE molecular weight is 421 g/mol. It is a positively charged surfactant with pKa at about 10–11 and the isoelectric point above pH 12. It is stable for more than two years at room temperature in a closed container. In an aqueous solution at 25 °C, its half-life decreases from more than one year at pH 4, to 57 days at pH 7, and 34 h at pH 9 [15], indicating its decomposition by base-catalysed hydrolysis. Thus, the combined effect of temperature with pH conditions markedly influences the hydrolysis of LAE to Nα-lauroyl–l-arginine (LAS) or arginine and lauric acid [15]. The commercially available food-grade LAE consists of not less than 85%, and not more than 95%, ethyl-*N*-α-lauroyl-l-arginate·HCl with the limits for contaminants from the synthesis set for: LAS (<3%); lauric acid (<5%); ethyl laurate (<3%); l-arginine·HCl (<1%) and ethyl arginate·2HCl (<1%) [3].

Adsorption properties of LAE at the water/air interface, its surface tension and critical micellisation concentration (CMC) have been investigated by many authors, but there are only a few reports where the surface tension isotherms can be found. Chi and Catchmark measured the surface tension isotherm of food-grade (≥98% purity) LAE and determined the CMC at 4.5 mM with the surface tension at CMC, σ_CMC_ = 26.4 mN/m [14]. Bai et al. used Mirenat-G containing 10.5% LAE in glycerol, measured the surface tension isotherm and established the CMC value at 0.1%wt (2.4 mM), and the σ_CMC_ around 26 mN/m [13]. In our recent work, we determined the surface tension isotherm of LAE surfactant (commercial name Mirenat-P/100) with about 90% surfactant content and obtained similar results [12]. Other authors presented CMC and σ_CMC_ values without reporting the isotherms. Their results are collected in Table 1.

The discrepancies between the results reported above can be attributed to the differences in LAE solution composition. They can contain surface-active residuals from LAE synthesis that are also the surfactant hydrolysis products. Namely, N_α_-lauroyl–l-arginine (LAS), that in neutral and mildly acidic conditions (above pH 5), is the amphoteric surfactant with a much lower solubility in aqueous media (<0.1 mM) than cationic LAE. Its minimal surface tension at the solubility limit is 43.8 mN/m [16]. Dodecanoic (Lauric) acid (DDA) has pK_a_ = 4.95. It is neutral at acidic pH and anionic at pH > 4.5. Consequently, each LAE solution is a mixed surfactant system, and the presence of additional surface-active components affects surface tension and CMC values.

The arguments presented above indicate that the thorough analysis of the surface activity of the well-defined LAE solutions still needs to be performed and the effect of surface-active contaminants evaluated. Moreover, the analysis of the dynamic surface tension and surface elasticity of LAE has never been conducted before. Our paper aimed to fill that void and to measure the surface tension isotherm of LAE of high purity (USP Reference Standard), determine the importance of different hydrolysis pathways by making quantum mechanical DFT calculations, and estimate the effect of the formation of LAE-DDA and LAE-LAS dimers that should exhibit very high surface activity. Considering the possibility of broad applications of LAE in biomedical, cosmetic and food processing areas, it is crucial to determine the mechanisms of its surface activity and aggregation properties.

## 2. Results and Discussion

We measured the surface tension of the freshly prepared LAE solution and determined its dependence on the surfactant concentration. The results are illustrated in Figure 1. The surface tension isotherm of LAE was compared with ones obtained for some model cationic and nonionic surfactants.

The onset of the surface activity of LAE was at the concentration 10^−4^ mol/dm^3^ and so at c.a. one order of magnitude lower concentration than of typical cationic surfactants with the same hydrocarbon chain length, dodecyl trimethylammonium bromide (C_12_TAB) [21] and dodecyl pyridinium chloride (C_12_PyCl) [22]. Simultaneously, that onset was at the concentration at least one order of magnitude higher than of nonionic surfactants such as nondissociated dodecanoic acid (C_11_COOH) [23], *n*-dodecyl-*β*-d-glucoside (C_12_Glu) [24] or *n*-dodecyl dimethyl phosphine oxide (C_12_PhospOx) [25]. The surface activity of LAE was the most similar to that observed for the solution of *N*,*N*,*N*-trimethyl-2-(dodecanoyloxy)ethane ammonium bromide (C_11_DMM) [26]. It was demonstrated that the surface activity of that surfactant resulted from the synergistic effect of adsorption of cationic surfactant—C_11_DMM and surface-active anion—dodecanoate that was the product of surfactant hydrolysis. C_11_DMM and dodecanoate can form electrostatically bound heterodimers with very high surface activity [26]. The CMC value of LAE was at the concentration 1.0–1.1 mmol/dm^3^, in agreement with the value reported in [18], lower than in other reports (see Table 1). The surface tension value at CMC (σ_CMC_) was 25 mN/m, much lower than for typical cationic surfactants, closer to σ_CMC_ for nonionic surfactants and characteristic for di-chain or Gemini surfactants [27,28]. Thus, such a low value of σ_CMC_ may support the idea of the formation of surface-active dimers, dodecanoate anion, that can be present in the solution at pH > 4.5 as the residual product of LAE synthesis or its hydrolysis. There are reproducible peculiarities of the isotherm shape at the concentration above 0.5 mM. The surface tension decreases but with a much lower slope to CMC.

Upon addition of salt, LAE behaves as a typical ionic surfactant, i.e., its surface tension drops due to the screening of electrostatic interaction between adsorbing surfactant molecules that contributes to the increase in surface activity, as illustrated in Figure 2a. After prolonged storage of the stock solution (over two weeks in 4 °C, pH 4.5), a significant change of the isotherm shape was observed, as shown in Figure 2b. It had a less steep slope, and the surface tension values were lower than for freshly prepared LAE solution for concentrations below 0.2 mM, and higher above that value. We hypothesised that the observed variation of the surface tension values resulted from changes in the surfactant solution composition due to LAE hydrolysis.

LAE can undergo hydrolysis via two paths. In the first path, the amide bond between the hydrophobic hydrocarbon chain and the hydrophilic headgroup can be hydrolysed, resulting in the l-arginine ethyl ester that is not a surface-active component, and surface-active dodecanoic (lauric) acid (see Figure 1) that above its pKa assumes the form of dodecanoate anion. In the second path, the ester bond linking side ethyl group hydrolyses, giving the surface-active N_α_-lauroyl–l-arginine and ethanol. We used the quantum chemical DFT computations to evaluate the energetics of base and acid-catalysed hydrolysis proceeding along these pathways. The schemes of reactions are depicted in the Appendix A and the results of computations are given in Table 2.

Considering the DFT computation results given in Table 2, we concluded that base-catalysed hydrolysis is irreversible; free energy of hydrolysis is equal to 15.6 kcal/mol for the first path and 24.2 kcal/mol for the second path. The entropic barrier, associated with the nucleophilic addition step, was much higher for the first path producing dodecanoate anion, than for the second, resulting in LAS formation. On the other hand, the acid-catalysed hydrolysis seemed to be not favourable at standard conditions (298 K, 1 atm). Although fresh LAE solution is mildly acidic, pH 5.15 at 4 × 10^−5^ M and pH 5 at 0.1 mM concentration (possibly because of the presence of lauric acid residues), due to the cationic charge of the surfactant, the base-catalysed hydrolysis can occur with a measurable rate. This occurs, in particular, at the water/air interface or at interfaces of surfactant micelles that are highly positively charged, whereby a local concentration of hydroxyl anions is increased. LAS was identified as the main product of LAE hydrolysis in simulated gastrointestinal conditions [29], and the LAS involving hydrolysis path was recognised as the first stage of the metabolic pathway for ethyl lauroyl arginate [30].

Hydrolysis products of both paths, bearing the hydrocarbon tail, dodecanoate anion or dodecanoic acid and LAS, are surface-active. Moreover, they can interact with LAE molecule forming heterodimers. In the solution, besides the hydrophobic effect of two hydrophobic tails, the LAE-dodecanoate heterodimer is bound by the electrostatic interaction of oppositely charged molecules. Moreover, due to three hydrogen bond donors of LAE, that heterodimer can be stabilised by hydrogen bonds. LAS is a zwitterionic molecule (at pH around 5); thus, its electrostatic interaction with LAE is weaker. On the other hand, due to 4 hydrogen bond donors and 6 acceptors, they can form hydrogen bounded heterodimers. Moreover, it was reported that guanidinium groups, despite their positive charge, could pair in salts solutions that may additionally contribute to the heterodimers’ stabilisation [31,32].

We used the DFT computations to evaluate the energetics of formation of the heterodimers LAE-dodecanoate anion and LAE-LAS. The optimised structures of those dimers are illustrated in Figure 3, and the results of computations are given in Table 3. They indicate the favourable formation of heterodimers with a stronger tendency to form LAE-LAS aggregates despite missing electrostatic attraction between molecules. The creation of LAE-LAS heterodimers can be followed by the formation of larger aggregates as the hydrolysis progresses. We observed that after a week of storage of 1 mM LAE solution, a white cloudy phase appeared with precipitated needle-like crystals, and the surface tension of the supernatant phase systematically increased. After 14 days, it reached 30 mN/m, about 5 mN/m higher than for fresh LAE solution, while pH of the solution decreased to 4.5.

We examined the existence and stability of the heterodimers at the air/water interface by molecular dynamics simulations. Figure 4 presents the representative snapshots illustrating the conformation of dimers at the interface.

Using the algorithm implemented in the YASARA Structure simulation package [33], we determined the distance between molecules forming heterodimers during the simulation run (300 ns) after the molecules appeared at the interface, and counted the number of hydrogen bonds. The results are illustrated in Appendix A. Despite the competition of surrounding water in the interfacial layer for hydrogen bonding, the intermolecular hydrogen bonds contributed to the formation of persistent dimers. The LAE-dodecanoate formed the heterodimer during 35% of the simulation run with the average number of bonds 1.71, while the average number of hydrogen bonds of both molecules with water was 9.8. Additionally, the heterodimer was stabilised by electrostatic interactions between cationic LAE and the dodecanoate anion. The LAE-LAS formed the heterodimer during 50% of the simulation run with the average number of intermolecular hydrogen bonds 1.96, while the average number of hydrogen bonds of both molecules with water was 14.6. The heterodimer can be additionally stabilised by the interactions of guanidinium groups not accounted for in the applied molecular dynamics force field [32]. The presence of the intermolecular hydrogen bonds between *n*-dodecyl-*β-*d-maltoside molecules at the air/water interface was recently demonstrated by molecular dynamics simulations and grazing-incidence X-ray (GIX) scattering and diffraction [34]. The molecular dynamics simulations result was in agreement with QM computations and indicated that LAE forms more stable heterodimers with LAS. It means that with the progress of hydrolysis, the amount of those dimers increases, which should be reflected in the change of the surface activity of the resulting surfactant mixture.

We attempted to model the LAE surface tension isotherms utilising the description of adsorption of mixtures of surface-active compounds based on the quasi-two dimensional electrolyte model of ionic surfactants adsorption [21,26]. Details of the model are given in the Appendix A. Considering the outcome of QM and MD simulations, we assumed that for the fresh solution, LAE was not hydrolysed but contained less than 0.5% molar of decanoic acid as a residual component from the synthesis. For the surfactant stored for more than two weeks, we assumed that the solution comprised a mixture of LAE and LAS. Since the formation of heterodimers was energetically favourable, we assumed that in both cases, the mixture contained monomeric LAE and LAE-dodecanoate anion or LAE-LAS dimers, respectively. As illustrated in Figure 2a, we obtained a satisfactory description of the experimental surface tension isotherms for the fresh LAE solution assuming 0.2% molar of dodecanoate anions, without and with the addition of NaCl. For the stored solution, a satisfactory description could be received assuming 18% molar of LAS resulting from the hydrolysis of the LAE stock solution during storage (cf. Figure 2b). The values of the best-fit parameters are collected in Appendix A. Since, with ageing, the composition of the mixture changes, fraction of LAS increases and pH decreases, the LAE-DDA neutral heterodimers are replaced with positive heterodimers LAE-LAS that result in the apparent CMC increase. Simultaneously, when pH decreases, fraction of LAS becomes protonated, cationic, and thus less surface-active with a lower tendency to form heterodimers with LAE.

We used the oscillating drop shape tensiometry to determine the dynamic interfacial properties and the surface dilational viscoelasticity of LAE solutions. The applied frequency of the drop oscillation was in the range of 0.01 and 0.2 Hz. Figure 5 presents the oscillations of the drop area (dashed line) and the corresponding changes of the surface tension for LAE concentration: A, 0.2 mM; B, 0.5 mM; C, 0.8 mM; D, 1 mM; and E, 1.5 mM, for the oscillation frequency 0.01 Hz and 0.1 Hz. For low surfactant concentration, the oscillation of the surface tension had a sinusoidal shape. At higher frequencies, the signal was noisy, presumably due to the slow relaxation of the interfacial layer composed of surfactant mixture. For the concentration 0.8 mM, the sinusoidal oscillations of the surface tension became distorted at the compression, with a large phase shift between the variations of drop area and surface tension. Even though the drop area continued to be compressed, the surface tension started to increase due to surfactant desorption. The distortion of the surface tension was more pronounced at higher frequencies and increased for the LAE concentration close to (1 mM) and above (1.5 mM) CMC.

Figure 6 illustrates the frequency dependence of the real and imaginary part of the dilational elasticity modulus determined for various concentrations of LAE solutions. The real part of the modulus increases with drop area oscillation frequency (ν) and can be successfully described by fitting the Lucassen-van den Tempel diffusional adsorption model valid for the surfactant concentration below CMC [35,36]:(1)$$\varepsilon =\varepsilon_{0}\frac{1+\xi +i\xi }{1+2\xi +2\xi^{2}},\;\xi =\sqrt{\frac{\nu_{D}}{2\nu }}$$
where: $\varepsilon_{0}$ is the Gibbs elasticity, $\nu_{D}$ is the characteristic frequency of the diffusion transport mechanism. Best fit parameters are collected in Table 4. On the other hand, the imaginary part of the dilational elasticity modulus can be described by that model only for low frequencies and the concentrations below 0.5 mM, which indicates that the surface elasticity of LAE solutions needs to be described by more complex models than diffusional adsorption of single surfactant [37]. The maximum of the real and imaginary part of surface dilational elasticity moduli was observed at 0.3 mM and they decreased with increasing surfactant concentration (cf. Appendix A). Above the CMC at higher frequencies, the surfactant layer seemed to be more viscous as the imaginary part of the modulus increased and exceeded the real one. This can be a consequence of shear effects and a nonlinear response for drop oscillation [36]; however, Loglio et al. attributed the increase in the phase shift, i.e., the imaginary part of the modulus, with the drop oscillation frequency, to the presence of surfactant mixture [38]. After storage for 14 days, as the surface tension at CMC (1 mM) increased by c.a. 5 mN/m, both modulus components also increased and surfactant layers seemed more elastic.

To evaluate the effect of the distortion of the surface tension oscillations from a sinusoidal shape, we calculated the dependence of the ratio of amplitudes of second to first harmonics in the Fourier spectrum F(Δσ)2F(Δσ)1 on LAE concentration. The results are illustrated in Figure 7. The onset of the nonlinear response of the LAE surface layer was around 0.5 mM; it increased up to CMC (1 mM) and levelled off. Therefore, low values of the elasticity modulus and the nonlinear response can be attributed to micellisation. Let us consider a single cycle of drop surface compression and decompression at the surfactant concentration at CMC. At the particular level of compression, the surface becomes oversaturated with the surfactant, desorption starts, and the desorbed surfactant is integrated into micelles; thus, the desorption rate is not attenuated by the local increase in monomeric surfactant concentration. Consequently, higher surface tension is observed with respect to sinusoidal dependence. Additionally, the heterodimers with higher area demand could be the first to be desorbed. Upon surface decompression, the desorption continues as long as the surface is oversaturated, then surfactant adsorption starts. Micelles play the role of the reservoir speeding up the adsorption, and the surface tension starts to decrease before the onset of drop surface compression. Consequently, the upper half of the surface tension oscillation cycle has a more regular shape. On the other hand, the irregular shape of the surface tension oscillations at LAE concentration 0.8 mM could also be attributed to the rearrangement of the surfactant layer due to the presence of heterodimers.

For low surfactant concentrations, the values of surface dilational modulus of LAE solutions were similar to those for DTAB [39]. However, for concentrations closer to CMC, they were much smaller, probably due to the neutralisation of the interfacial layer due to the presence of LAE-dodecanoate dimers. A similar dependence of the elasticity modulus on the surfactant concentration was observed for nonionic surfactant dodecyl dimethyl phosphine oxide [40]. The maximum was observed at a concentration about 10 times lower than CMC, a sharp drop at the CMC, and the constant value above. We found the increase in the phase shift of the surface tension variations with the drop oscillation frequency as suggested by Loglio et al. for surfactant mixtures [38]. The above observation is consistent with our claim that LAE solution should be considered as the mixed surfactant system consisting of monomeric LAE molecules and its heterodimers with dodecanoate anions or LAS.

To substantiate our hypothesis concerning the progressing hydrolysis of LAE during its storage as an aqueous solution, we compared the infrared absorption spectra of the freshly prepared solution of LAE, and after storing the stock solution of 0.2 wt%. concentration for three weeks at room temperature. Additionally, we measured spectra of the freshly prepared 1.3 × 10^−3^ M solution of lauroyl arginine (LAS). To assign the frequencies of the vibration bands, we determined the theoretical spectra of LAE, LAS and LAE-LAS heterodimer in the IR region by the DFT computations. The results are illustrated in Appendix A.

After storing LAE solution, we observed precipitate in the form of needle-like crystals as shown in the microscopic image in Figure 8. Figure 9 illustrates the comparison of the IR spectra of freshly prepared LAE and LAS solution in the range 1800–900 cm^−1^ with ones obtained for the “stored” LAE solution taken from the precipitate and supernatant. Comparing the spectra illustrated in Figure 9 and Appendix A, we concluded that precipitate contained both LAE and LAS, and the formation of heterodimers was followed by the growth of needle-like crystallites. On the other hand, the supernatant phase consisted of a mixture of LAE and LAS in the protonated (as pH of the solution decreased with time to 4.5) form with a higher ratio of the latter. In both cases, we observed a decrease in the intensity and shift of the band associated with carbonyl of the ester group to lower wavenumbers and the appearance of the bands characteristic for the carboxylic group in the ionic (1400 cm^−1^) and noionic (1200 cm^−1^).

## 3. Materials and Methods

### 3.1. Materials

Ethyl lauroyl arginate, United States Pharmacopeia analytical standard (declared purity 99%) and lauroyl arginine hydrochloride (LAS), (United States Pharmacopeia reference standards) were purchased from Merck, Warsaw, Poland). Sodium chloride (99%) was acquired from Sigma-Aldrich, Poznan, Poland and calcinated in 650 °C for eight hours before use. Laboratory glassware was cleaned with Helmanex solution, sulfuric acid, and deionised water. LAE was stored at 4 °C and protected from light. Before measurements, the stock solution was prepared in deionised cold water (4 °C, 20 MΩ) and then diluted to the appropriate concentration. Stock solution and dilutions were used within one day if not described otherwise.

### 3.2. Surface Tension and Elasticity

Surface tension and elasticity were measured by the pendant drop technique with the Sinterface PAT-1M (Sinterface, Berlin, Germany) tensiometer. Steel capillary of 2 mm diameter was cleaned carefully before each measurement. A drop of solution (11 µL) was created and kept in a thermostated chamber (22 °C) until reaching equilibrium surface tension. If not described otherwise, the surface tension and elasticity measurements were performed within one day after solutions’ preparation. For the dynamic surface tension measurements, the drop profile coordinates were recorded every second and fitted with the Young-Laplace equation to calculate the surface tension. The precision of measurements was 0.1 mN/m.

Surface elasticity modulus was determined after reaching surface tension equilibrium by imposing drop oscillations of less than 10 percent of its volume. Raw data of the surface tension variations in response to periodic drop surface area changes were smoothed applying Loess smoother. Then Fourier transform was calculated and the surface dilatational modulus was determined as the complex number [37]:(2)$$\varepsilon =\varepsilon_{r}+i\varepsilon_{i}=A_{0}\frac{\Delta \sigma_{1}}{\Delta A_{1}}$$
where: $\varepsilon_{r},\;\varepsilon_{i}$ are the real and imaginary part of the dilational elasticity modulus, *A*_0_ is the average area of the drop, $\Delta A_{1}$ and $\Delta \sigma_{1}$ are the principal Fourier components of the area and surface tension variations that correspond to the frequency of drop oscillations. All calculations were performed with the Mathcad (Parametric Technology Corporation, Needham, MA, USA) script.

### 3.3. Quantum Chemical DFT Computations

The quantum mechanics computations were performed using density functional theory (DFT) with wB97XD functional, which includes corrections for dispersion and long-range interactions, using a 6-31G+(d,p) basis set [41]. Solvation effects (water) were accounted for applying the SMD variation of the Polarisable Continuum Model [42]. To evaluate the relative rates of hydrolysis, the molecular structures of LAE, all its hydrolysis products and transition states were minimised, and energy, enthalpy, and free energy of a particular structure were determined. Then the respective energies, enthalpies and free energies of hydrolysis and the height of energetic barriers were calculated for basic and acidic hydrolysis in standard conditions (298 K, 1 atm).

The energy of dimerisation of LAE with surface-active hydrolysis products, dodecanoate anion and LAS was determined as follows. We placed two molecules with optimised geometries with parallel oriented hydrophobic chains and random position and orientation of the headgroup. The optimising procedure was run until the convergence was achieved, and the energy, enthalpy and free energy of the heterodimer were obtained. The procedure was repeated three times for different initial positions of headgroups and the conformation with the lowest energy was selected. The energy, enthalpy, and free energy of dimerisation were calculated according to: $\Delta E_{dimerization}=E_{dimer}-E_{LAE}-E_{LAS/dodecanoate}$. All DFT calculations were carried out using the Gaussian 09 program [43].

### 3.4. Molecular Dynamics Calculations

The optimised structures of heterodimers obtained in the DFT computations were imported to the YASARA Structure Molecular Dynamics Software [33], placed in the simulation box with the size of 5 × 5 × 5 nm filled with water molecules (TIP3P, density 1 g/dm^3^). The simulation was run for 20 ns using AMBER 14 force field [44] to equilibrate the system. Then it was continued for 300 ns and the positions of the investigated molecules were monitored. For the simulation of molecules at interface, after equilibration, the simulation box was extended in z coordinate to 15 nm to obtain a water slab with two interfaces. While the simulation was running, the transfer of molecules to one of the interfaces was observed. Then the simulation was continued for 300 ns with the recording of molecules’ positions. The distance between LAE and LAS or dodecanoate anion was monitored and the number of hydrogen bonds between heterodimer forming molecules and the number of hydrogen bonds between these molecules and water was determined every 0.1 ns using the algorithm implemented in the YASARA Structure Software [33] and AMBER force field parameters.

### 3.5. Infrared Spectroscopy

The infrared spectra were collected using IR microscope Nicolet iN10 (Thermo Scientific™ part of Thermo Fisher Scientific, Madison, WI, USA) with high sensitivity, LN-cooled MCT detector by the reflection mode measurements within the spectral range from 4000 cm^−1^ to 675 cm^−1^. After recording the background, aqueous solutions of investigated compounds were drop-casted on a gold layer sputtered on glass plates. Spectra (128 scans during 45 s with high resolution 4 cm^−1^) were recorded after water evaporation. The fully automated adjustable aperture for measuring field extraction was 150 µm × 150 µm. After measurements, the automatic correction, namely, atmospheric, baseline subtraction and scale normalisation, was applied, and the averaged spectra were created from at least three spectra collected from different places of a sample.

## 4. Conclusions

Considering ethyl lauroyl arginate (LAE) applications in biomedical, cosmetic and food processing areas, it is crucial to define the mechanisms of its surface activity and aggregation properties. We determined the surface tension isotherm and surface dilational viscoelastic moduli of LAE solutions with analytical standard purity (>99%). We established that the surface activity was in-between one for cationic and nonionic surfactants with the same length of the hydrophobic tail. LAE solution can contain the residues from the synthesis that are surface-active, namely lauric acid. Moreover, during storage, LAE undergoes base-catalysed hydrolysis that is enhanced at a positively charged interface or surface of micelles of LAE cationic surfactant. The preferred hydrolysis pathway of that process leads to Nα-lauroyl–l-arginine (LAS), a zwitterionic surface-active component, that was supported by the IR spectroscopy analysis. Therefore, every LAE solution is a multicomponent system.

We used quantum mechanical DFT computations to determine the energetics of the hydrolysis paths and evaluated the possibility of the formation of highly surface-active heterodimers, LAE-dodecanoate anion or LAE-LAS. We used molecular dynamics simulations to determine the stability of those dimers linked by electrostatic interactions and hydrogen bonds (LAE-dodecanoate anion) or hydrogen bonds and stacking of guanidinium groups (LAE-LAS). We applied the model of surfactant mixtures adsorption to successfully describe the experimental surface tension isotherms assuming the presence of 0.2% of dodecanoic acid in fresh LAE solution and 18% of LAS after its hydrolysis during prolonged storage. The surface dilational modulus measurements by the oscillation drop method revealed values of surface elasticity moduli between ones for ionic and nonionic surfactants. The nonlinear response of the surface tension for the drop oscillations that could be observed for LAE concentration close to, and above, CMC was attributed to the presence of micelles and the reorganisation of the interfacial surfactant layer.

## Figures and Tables

**Figure 1 molecules-26-05894-f001:**
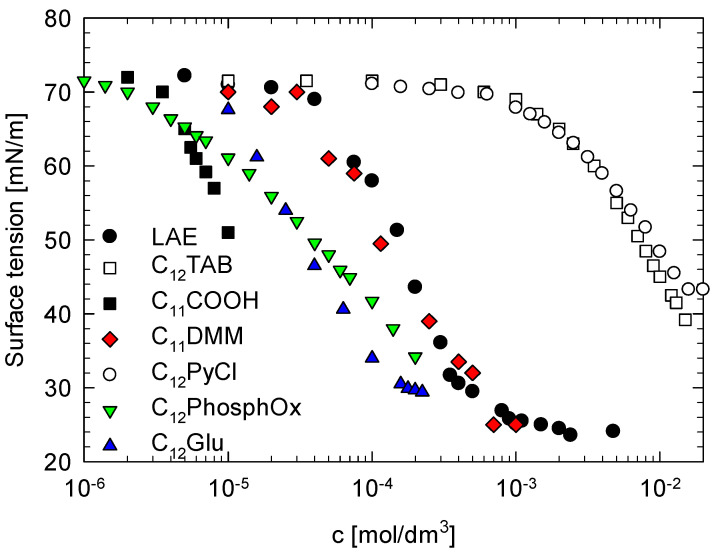
The comparison of the surface tension isotherm of LAE with ones obtained for some model cationic and nonionic surfactants: dodecyl trimethylammonium bromide (C_12_TAB), dodecyl pyridinium chloride (C_12_PyCl), (C_11_COOH), *n*-dodecyl-*β*-d-glucoside (C_12_Glu), *n*-dodecyl dimethyl phosphine oxide (C_12_PhospOx) and *N*,*N*,*N*-trimethyl-2-(dodecanoyloxy)ethane ammonium bromide (C_11_DMM).

**Figure 2 molecules-26-05894-f002:**
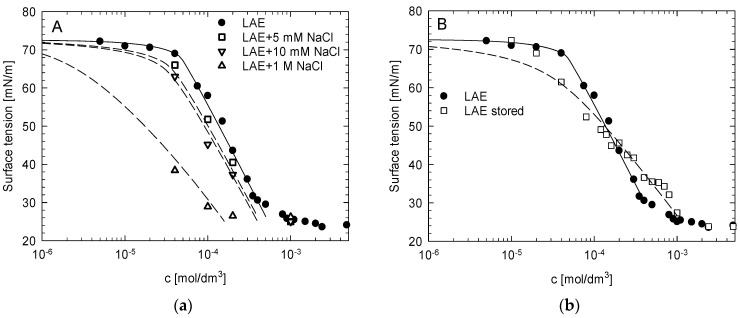
(**a**) The change of the dependence of surface tension on the concentration of LAE solution upon addition of salt (NaCl); (**b**) the comparison of LAE surface tension isotherms for freshly prepared and “stored” solution. Lines represent fits to the model of LAE adsorption accounting for the formation of LAE-dodecanoate or LAE-LAS heterodimers.

**Figure 3 molecules-26-05894-f003:**
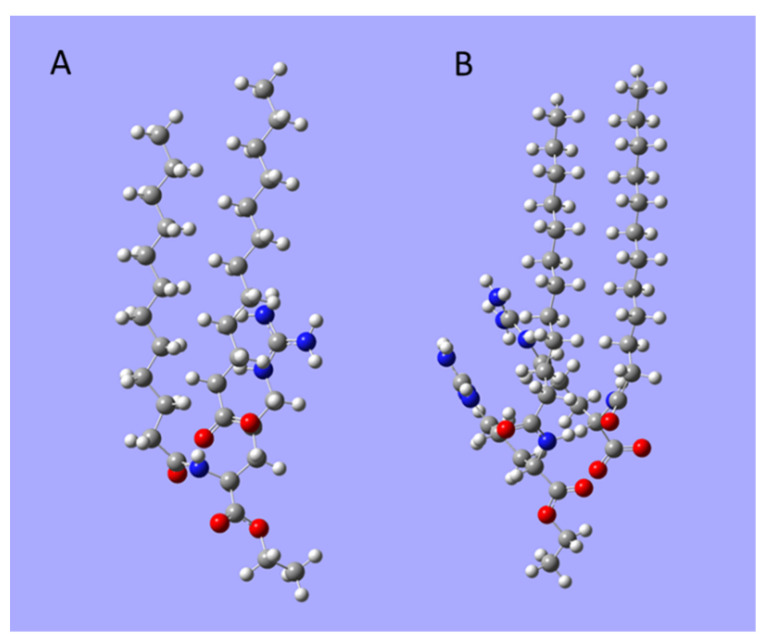
The optimised structures of (**A**) LAE-dodecanoate, (**B**) LAE-LAS heterodimers. Grey, carbon; white, hydrogen; red, oxygen; blue, nitrogen.

**Figure 4 molecules-26-05894-f004:**
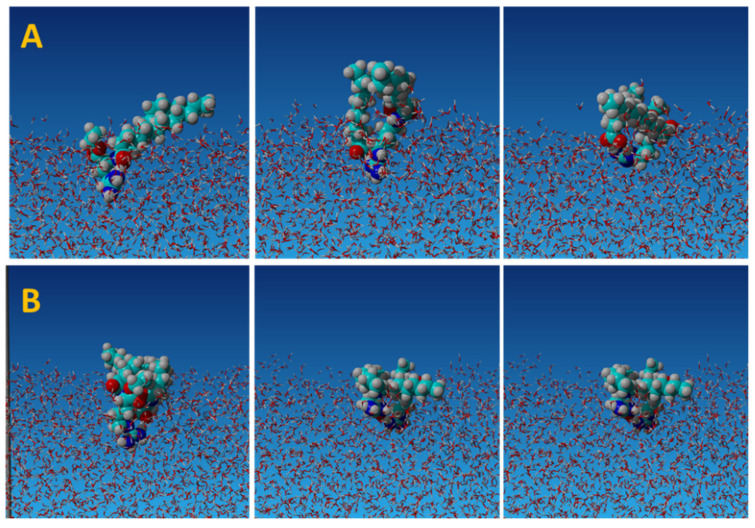
The examples of snapshots of molecular dynamics simulations of heterodimers at air/water interface. (**A**) LAE–dodecanoate; (**B**) LAE-LAS. Light blue, carbon; white, hydrogen; red, oxygen; blue, nitrogen.

**Figure 5 molecules-26-05894-f005:**
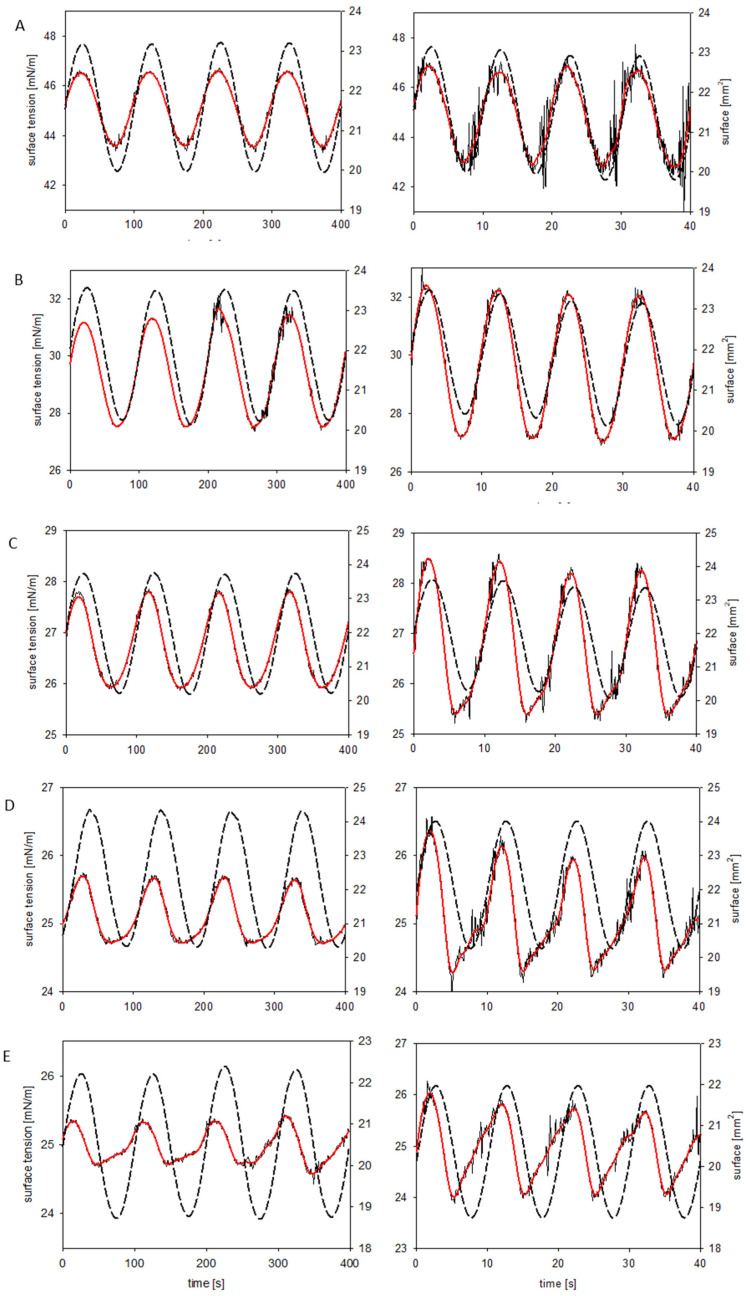
Oscillations of the drop area (dashed line) and the corresponding changes of the surface tension. Red line: the surface tension after Loess smoothing. LAE concentration: (**A**) 0.2 mM; (**B**) 0.5 mM; (**C**) 0.8 mM; (**D**) 1 mM; (**E**) 1.5 mM. Frequency: left, 0.01 Hz; right, 0.1 Hz.

**Figure 6 molecules-26-05894-f006:**
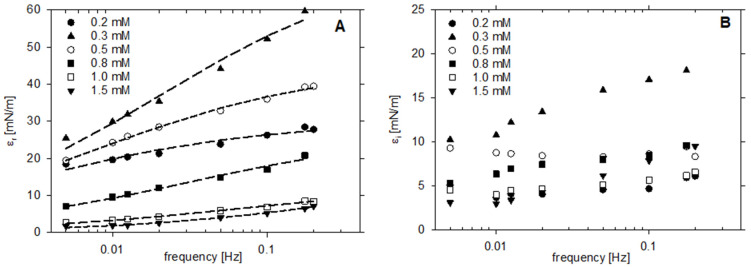
The dependence of (**A**) real, and (**B**) imaginary, parts of the dilational elasticity modulus on the drop surface oscillation frequency for various concentrations of LAE solutions. Dashed lines: fits of the Lucassen-van den Tempel diffusional adsorption model.

**Figure 7 molecules-26-05894-f007:**
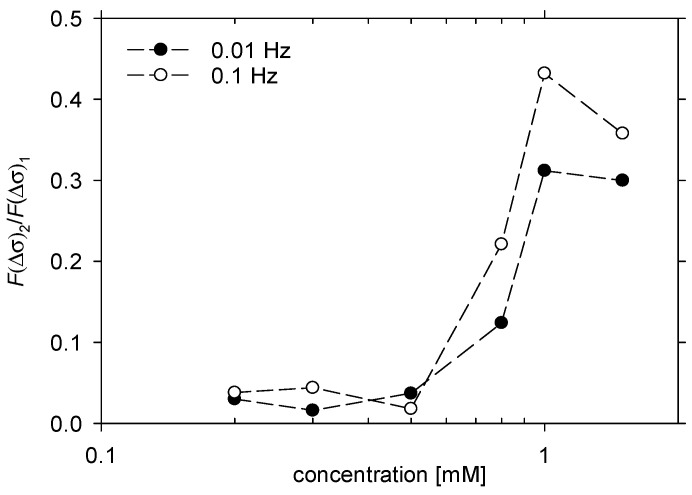
The dependence of the ratio of second to first harmonics in the Fourier spectrum of surface tension oscillations on LAE concentration for frequency of 0.01 Hz and 0.1 Hz. Lines were drawn to guide the eye.

**Figure 8 molecules-26-05894-f008:**
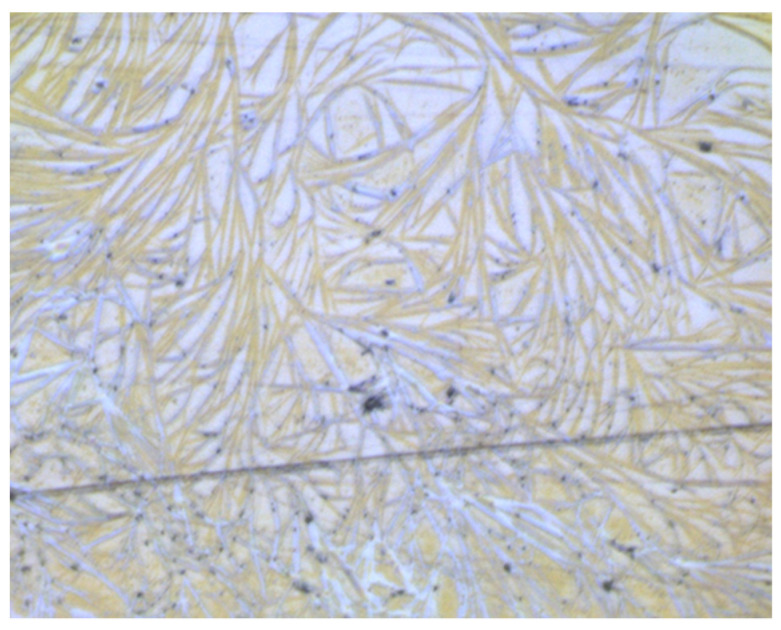
Microscopic image of a precipitate from “stored” LAE solution. Image size 500 × 400 µm.

**Figure 9 molecules-26-05894-f009:**
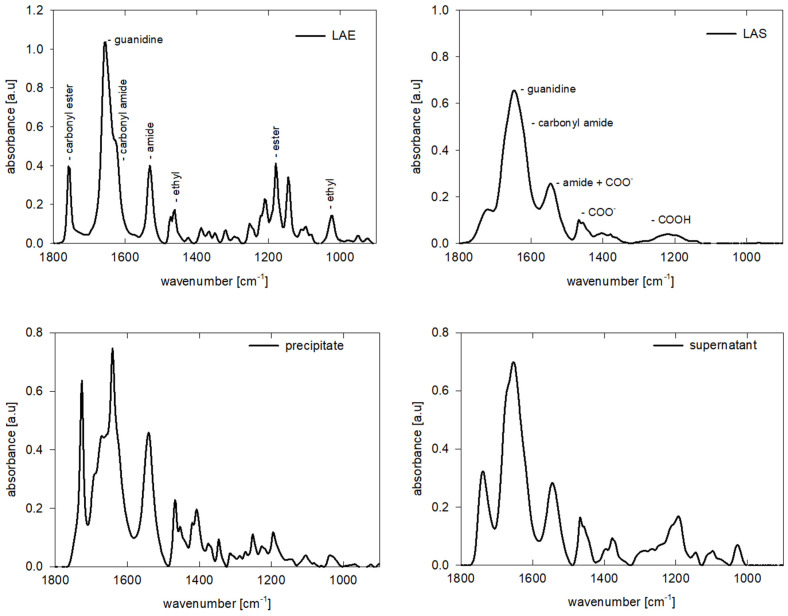
IR spectra of fresh LAE and LAS solutions and of “stored” LAE, precipitate and supernatant phase.

**Table 1 molecules-26-05894-t001:** The results of characterization of CMC of LAE solution by various authors.

Scheme	LAE Content	CMC (mM)	σ_CMC_ (mN/m)	Method	Reference
Mirenat-CF	10.5% in propylene glycol	4.9		Isothermal titration calorimetry	[2]
			-		
Mirenat-CF	10.5% in propylene glycol	4.5		Isothermal titration calorimetry	[9]
			-		
LAE Synthesized	-	6.0	31.8	Ring tensiometer	[16]
LAE (LAMIRSA)	85–95%	2.4	25.4	Ring tensiometer	[17]
LAE.HCl (local supplier)	-	0.9	25.5	Conductivity, ring tensiometer	[18]
LAE synthesized	>95%	6.2	30.2	Ring tensiometer	[19,20]

**Table 2 molecules-26-05894-t002:** Energies, enthalpies and free energies of LAE hydrolysis reactions ΔE_h_. and their transition states ΔE_t_.

Hydrolysis Path		Base Catalysed		Acid Catalysed	
	Data	ΔE_t_ [kcal/mol]	ΔE_h_ [kcal/mol]	ΔE_t_ [kcal/mol]	ΔE_h_ [kcal/mol]
LAE → l-arginine ethyl ester + dodecanoate/dodecanoic acid	energy	7.0	−10.3	20.7	11.2
	enthalpy	6.4	−10.3	20.1	11.2
	free energy	17.5	−15.6	31.3	9.0
LAE → LAS + ethanol	energy	−4.3	−21.6	12.8	5.7
	enthalpy	−4.3	−21.6	12.2	5.7
	free energy	4.9	−24.2	23.8	4.4

**Table 3 molecules-26-05894-t003:** Energies, enthalpies and free energies of formation of LAE heterodimers.

Dimerization	Energy [kcal/mol]	Enthalpy [kcal/mol]	Free Energy [kcal/mol]
LAE-dodecanoate	−20.2	−21.0	−1.6
LAE-LAS	−29.4	−29.9	−6.5

**Table 4 molecules-26-05894-t004:** Best fit parameters of the Lucassen-van der Tempel model to the real part of surface dilational elasticity modulus of LAE solutions.

Concentration [mM]	$\varepsilon_{0}\;[\mathrm{mN}/m]$	$\nu_{D}\;[\mathrm{Hz}]$
0.2	30.1	0.0032
0.3	73.9	0.02
0.5	45.3	0.0081
0.8	26.9	0.03
1.0	12.6	0.06

## Data Availability

The data presented in this study are openly available from the authors upon reasonable request.

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
