# Peer review of "Ethyl Lauroyl Arginate, an Inherently Multicomponent Surfactant System"

_molecules, 2021, doi:10.3390/molecules26195894_

Round 1

Reviewer 1 Report

In this work the surface tension and dilational elasticity behavior of Ethyl lauroyl arginate solutions are studied. The formation of heterodimers is proposed and supported by quantum mechanical density functional theory and molecular dynamic simulations

The paper is well written. It deserves a better revision on the formation of such dimers to support the explanation given.

Specific comments

Figure 1 line 81 place de full name of each surfactant on the figure description

Line 93 heterodimers with very high surface activity. Insert a reference

Line 198 the high phase shift at high frequencies could be due to shear effects. At higher frequencies phase angle should be lower, not higher, hence an explanation its desirable (or is it a measurement man made or equipment error?). Please revise carefully the following publication

Zamora, J. M., Marquez, R., Forgiarini, A. M., Langevin, D., & Salager, J. L. (2018). Interfacial rheology of low interfacial tension systems using a new oscillating spinning drop method. Journal of colloid and interface science, 519, 27-37.  https://doi.org/10.1016/j.jcis.2018.02.015

Line 202 correct to Figure 6

Line 213 this further adds to the possibility of the appearance of shear effects that makes the response more viscous

Line 221 LVDT model is valid at concentrations only below cmc. Above cmc the model is not appropriate. See the reference above https://doi.org/10.1016/j.jcis.2018.02.015

Best regards

Reviewer 2 Report

The article by Agnieszka Czakaj, Marcel Krzan and Warszyński describes their findings related to the surfactant Ethyl lauroyl arginate. Their main observations are the evolution of the surface tension isotherms of aqueous surfactant solutions and their mixtures with sodium chloride as well as the measurement of oscillatory dynamic surface tension using the bubble method.

The authors explain their findings by using Quantum chemical DFT computations and Molecular Dynamics. From the first they determine which degradation products should be present and from the second they determine the like hood of dimer formation.

In my opinion the subject of investigation may be of interest but the research is not yet mature.

The use of Quantum chemical DFT computations to determine which degradation products could be present in the system is valid, but it should be accompanied by chemical determination of the species present in the system after time evolution. From the authors data, it seems that the degradation occurs quite fast in basic conditions and slower in acidic conditions. “its half-life decreases from more than one year at pH 4 to 57 days at pH 7 and 34 h at pH 9 [15], indicating its decomposition by base-catalysed hydrolysis” Reference 15 seems to be wrong and the information seems to be in reference 16. Apparently this data was obtained by the producer, although from the reference I cannot see who is giving this data, maybe some independent research should confirm this point. Doing this also the identification and quantification of the degradation should be possible and contrasted with the theoretical findings exposed here.

The use of molecular dynamics to understand the possible formation of dimers at the surface seems appropriate, but depends on the correct identification of the species at the surface.

Because the authors observe time evolution of the aqueous samples, they should clearly state the age of each sample at measurement and, also the pH at which the sample has been kept. Are the samples buffered? If not, which is their pH?

It is a bit confusing the exact nature of the surfactant used because in the introduction so much emphasis is made on non-pure surfactants. For instance, in table 1 all samples are mixtures, except, probably for the synthesized one (ref.16). It is quite notable the difference in CMC obtained in the literature in ref. 16 (6 mM) compared with the present case (1 mM). It is usually found in the literature that impurities decrease CMC. The authors should comment on this. Also the apparent increase of CMC with aging seems to be contradictory with the literature trends.

The comparison with other surfactants could be fine, but, maybe they should also refer to closerly related surfactants like Methyl lauroyl arginate which seems to have a CMC in the range 4.5 to 6 mM (L. Perez et al. J. Phys. Chem. B 2007, 111, 11379-11387) or other lauroyl derivatives of other bulky headgroups (because we should bear in mind that the polar head contains up to 8 carbon atoms).

“The above observation is consistent with our claim that LAE solution should be considered as the mixed surfactant system consisting of monomeric LAE molecules and its heterodimers with dodecanoate anions or LAS.” This is the last sentence of the results and discussion section. It may be because I’m not an expert in surface rheology, but from the authors discussion I do not see very clearly the conclusion. Maybe the authors should explain this point better.

Also, because of my ignorance on surface rheology, it surprises me that the elastic component increases with surfactant concentration to decrease after. Is this the normal surfactant behavior or it is only normal for mixtures? Looking at the references given, I did not find an easy response.

There are a few mistakes that should be corrected. The numbering of the figures in the text seems to be wrong. Line 193 it seems the authors refer to figure 5, and in line 202 to figure 6. Line 246 refers to part a and a of the same figure.

Therefore, I suggest the authors to do some analytical determinations to support their theoretical findings. Also the discussion should be improved to better convince the reader of their views.

Round 2

Reviewer 2 Report

From the authors replays I'm half-way satisfied. The authors have corrected the most obvious mistakes like figure numbering and have added some information that was missing. The authors do not agree, however, with the reviewer demand of experimental evidence of the degradation products. I'm doubtful about whether the information provided by the authors is enough to support their hypothesis.

The authors have answered to some of the questions raised. However, in the revised version, is is not clear what changes have been made. The authors delete complete paragraphs but only a number or so has changed. Only the insertions are clearly identified.

Therefore, the authors have not answered my main concern in a clear way. I can not support poblication, although I acknowledge that the manuscript has improved somewhat.
